# POMC neurons expressing leptin receptors coordinate metabolic responses to fasting via suppression of leptin levels

Alexandre Caron[1†], Heather M Dungan Lemko[2†], Carlos M Castorena[1], Teppei Fujikawa[3], Syann Lee[1], Caleb C Lord[1], Newaz Ahmed[1], Charlotte E Lee[1], William L Holland[4], Chen Liu[1,5], Joel K Elmquist[1,6*]

[1]Division of Hypothalamic Research, Department of Internal Medicine, University of Texas Southwestern Medical Center, Dallas, United States; [2]Howard Community College, Columbia, United States; [3]Department of Cellular and Integrative Physiology, UT Health San Antonio, San Antonio, United States; [4]Touchstone Diabetes Center, Department of Internal Medicine, University of Texas Southwestern Medical Center, Dallas, United States; [5]Department of Neuroscience, University of Texas Southwestern Medical Center, Dallas, United States; [6]Department of Pharmacology, University of Texas Southwestern Medical Center, Dallas, United States

*For correspondence:
Joel.Elmquist@utsouthwestern.edu

[†]These authors contributed equally to this work

**Abstract** Leptin is critical for energy balance, glucose homeostasis, and for metabolic and neuroendocrine adaptations to starvation. A prevalent model predicts that leptin's actions are mediated through pro-opiomelanocortin (POMC) neurons that express leptin receptors (LEPRs). However, previous studies have used prenatal genetic manipulations, which may be subject to developmental compensation. Here, we tested the direct contribution of POMC neurons expressing LEPRs in regulating energy balance, glucose homeostasis and leptin secretion during fasting using a spatiotemporally controlled *Lepr* expression mouse model. We report a dissociation between leptin's effects on glucose homeostasis versus energy balance in POMC neurons. We show that these neurons are dispensable for regulating food intake, but are required for coordinating hepatic glucose production and for the fasting-induced fall in leptin levels, independent of changes in fat mass. We also identify a role for sympathetic nervous system regulation of the inhibitory adrenergic receptor (ADRA2A) in regulating leptin production. Collectively, our findings highlight a previously unrecognized role of POMC neurons in regulating leptin levels.

DOI: https://doi.org/10.7554/eLife.33710.001

## Introduction

Pro-opiomelanocortin (POMC) neurons of the arcuate nucleus of the hypothalamus (ARC) are critical regulators of energy balance and glucose homeostasis (*Mercer et al., 2013*; *Gautron et al., 2015*). These neurons consist of a heterogeneous population with respect to neurotransmitters used and the receptors expressed (*Hentges et al., 2009*; *Williams et al., 2010*; *Lam et al., 2017*). Electrophysiology and immunohistochemistry studies have established that ~30% of hypothalamic POMC neurons are responsive to leptin (*Cheung et al., 1997*; *Ernst et al., 2009*; *Williams et al., 2010*). Given the role of POMC neurons and leptin in metabolism, a conventional model indicates that a subset of POMC cells that expresses the leptin receptor (LEPR) are mediating the metabolic actions of leptin (*Cheung et al., 1997*; *Balthasar et al., 2005*). This idea was supported by early

observations that prenatal manipulations of LEPR-expressing POMC neurons mildly affect body weight (*Münzberg et al., 2003*; *Balthasar et al., 2004*; *Huo et al., 2009*; *Berglund et al., 2012*; *Huang et al., 2012*; *Mercer et al., 2013*). However, POMC neurons share developmental origins with other cell types, including subsets of NPY/AgRP neurons (*Padilla et al., 2010*; *Lam et al., 2017*). As such, it is possible that developmental compensation, or *Lepr* deletion from non-POMC neurons, are behind the phenotypes observed with conventional transgenic models (*Bouret et al., 2004*; *Lam et al., 2017*). In addition, although it was repeatedly suggested that leptin's anorexigenic effects act through non-ARC POMC neurons (*Myers et al., 2009*; *Berglund et al., 2012*; *Berglund et al., 2013*), the direct contribution of LEPR-expressing POMC neurons on glucose homeostasis has been difficult to dissect due to inevitable alterations of fat mass resulting from pre-natal deletions. As such, dissociating the pathways involved in leptin's and melanocortin's effects on adiposity versus glucose homeostasis is key for the development of anti-obesity and anti-diabetes therapies.

The activity and expression of POMC is highly dependent on energy status (*Mizuno et al., 1998*). During obesity, there is an energy surplus and POMC levels are elevated (*Schwartz et al., 1997*; *Cowley et al., 2001*). Inversely, during a state of negative energy balance, such as fasting, POMC expression is decreased (*Mizuno et al., 1998*). Because POMC deficiency causes severe obesity, tremendous efforts have been made to understand a causative role of the POMC neurons in the pathophysiology of both syndromic and diet-induced obesity (*Krude et al., 1998*; *Enriori et al., 2007*). However, relatively little is known about the function of these neurons in the context of low energy levels, despite early suggestions that the effect of fasting to reduce POMC is physiologically relevant (*Mizuno et al., 1998*). In addition, fasting leads to a rapid fall in circulating leptin levels that is out of proportion to the loss in fat mass (*Becker et al., 1995*; *Moinat et al., 1995*; *Saladin et al., 1995*; *Ahima et al., 1996*; *Flier, 1998*; *Ahima et al., 1999*). Despite early suggestions that the fall in leptin represent a central physiologic response to fasting required for metabolic adaptations to low energy states, the mechanisms behind fasting-induced reductions in leptin are unknown (*Ahima et al., 1996*; *Flier, 1998*; *Ahima et al., 1999*; *Flier and Maratos-Flier, 2017*). Paradoxically, LEPR-null animals do not experience a decrease in leptin levels with fasting, suggesting that LEPRs themselves are required for the starvation-induced fall in leptin (*Hardie et al., 1996*). Together, these observations indicate that neurons expressing LEPRs might play a role in repressing plasma leptin levels during starvation. However, the actual contribution of LEPR-expressing POMC neurons in regulating leptin secretion is unknown.

One way the CNS may regulate leptin is through altering activity of adrenergic receptors expressed by adipocytes. Acute activation of the sympathetic nervous system reduces leptin gene expression and leptin production through a β3-adrenoceptor (ADRB3)-dependent mechanism (*Moinat et al., 1995*; *Gettys et al., 1996*; *Giacobino, 1996*; *Mantzoros et al., 1996*; *Trayhurn et al., 1996*; *Deng et al., 1997*; *Trayhurn et al., 1998*; *Caron et al., 2018*). In addition, forcing the expression of human α2-adrenoreceptor (ADRA2) in mouse adipose tissue results in elevated leptin (*Valet et al., 2000*), suggesting that the ADRA2/ADRB3 balance in adipocytes is critical for leptin regulation. These observations suggest that leptin could regulate its own expression through a negative feedback loop from the brain to the adipose tissue. However, the central pathways and the mechanisms underlying these actions are yet to be fully characterized.

Here, we report that a subset of POMC neurons that express LEPRs directly control glucose homeostasis and are necessary to regulate leptin synthesis, independent of changes in fat mass. We used a tamoxifen-inducible *Pomc*^CreERt2 transgenic mouse model to generate mice in which *Lepr* expression is spatiotemporally-controlled in a neuron-specific fashion. Within one week of deleting LEPRs from POMC neurons in adult mice, hepatic glucose production was impaired, while body weight, food intake, and energy expenditure were unaltered. In addition, mice with adult deletion of LEPRs in POMC neurons showed an impairment in the fasting-induced fall in leptin levels. We also identified an important role for adipose tissue ADRA2A in regulating leptin synthesis. Our results support a model predicting that LEPR-expressing POMC neurons coordinate metabolic responses to fasting via suppression of leptin levels.

## Results

### LEPR-expressing POMC neurons are required for normal liver insulin sensitivity in adult mice

The use of conventional prenatal $Pomc^{Cre}$ models was key in deciphering the contribution of many receptors and pathways in glucose and energy homeostasis (*Hill et al., 2010*; *Xu et al., 2010*; *Berglund et al., 2012*; *Caron et al., 2016*). However, it is now appreciated that prenatal manipulations may lead to compensatory events during development (*Padilla et al., 2010*; *Bouret et al., 2004*)). Importantly, there is a subpopulation of cells that express $Pomc^{Cre}$ during development, but do not express POMC in adults (*Padilla et al., 2010*). To circumvent these issues, we used a tamoxifen-inducible $Pomc^{CreERt2}$ transgenic mouse model (*Berglund et al., 2013*) to generate $Pomc^{CreERt2}::Lepr^{flox/flox}$ mice in which $Lepr$ expression is spatiotemporally controlled in a neuron-specific fashion. We first assessed the impact of adult deletion of LEPR-expressing POMC neurons on glucose homeostasis. Fed and fasting glycemia were not different before, or one week after, injection of tamoxifen, indicating that the drug per se, did not impair glucose levels (*Figure 1A*). However, adult ablation of LEPRs from POMC neurons resulted in significantly higher fasting glycemia as early as two weeks post-deletion, while fed glycemia was greater at three weeks (*Figure 1A*). This effect was sustained for the entire experimental period. Fed and fasting insulin and glucagon levels were not different between groups (*Figure 1B–C*). Although no changes in glycemia were detectable in the first week, insulin response was already substantially impaired, as assessed by an insulin tolerance test (*Figure 1D–E*). We did not observe any difference in glycemia following a glucagon stimulation test (*Figure 1—figure supplement 2*).

We further explored the impact of deleting LEPRs in adult POMC neurons on systemic glucose metabolism by performing hypersulinemic-euglycemic clamp assays one week after the deletion in an independent cohort of animals. The glucose infusion rate needed to maintain euglycemia ($119.3 \pm 3.9$ vs $122.0 \pm 8.2$ mg/dl) was significantly decreased in knock-out animals (*Figure 1F*), further demonstrating whole-body insulin resistance. Importantly, glucose disposal was unaltered, but insulin-induced suppression of hepatic glucose production was drastically impaired in the clamped state (*Figure 1G–H*). Moreover, the ability of insulin to suppress lipolysis during the clamped state was unaltered, suggesting that insulin resistance occurred specifically in the liver (*Figure 1I*). Deletion of LEPRs in POMC neurons in adult mice did not affect fed or fasting levels of NEFA and triglycerides (data not shown), again suggesting that impaired liver insulin sensitivity, but presumably not impaired insulin secretion, contributes to systemic insulin resistance. Altogether, these data demonstrate that LEPR-expressing POMC neurons directly regulate liver metabolism in adult mice. This is in agreement with previous findings (*Hill et al., 2010*; *Berglund et al., 2012*) in which LEPRs were deleted during development. We found that insulin resistance can be detected one week post-deletion (*Figure 1D–I*), however blood glucose levels did not rise until two weeks post-deletion (*Figure 1A*). These findings suggest that deletion of LEPRs in adult POMC neurons impairs liver insulin sensitivity, and the resulting hepatic insulin resistance leads to the development of hyperglycemia.

### LEPR-expressing POMC neurons are dispensable for the regulation of energy balance in adult mice

It has generally been assumed that LEPR-expressing POMC neurons are important for feeding and weight regulation (*Cheung et al., 1997*; *Balthasar et al., 2005*), despite evidence that other subsets of POMC neurons are more likely to regulate energy balance (*Huo et al., 2009*; *Berglund et al., 2013*). Because prenatal deletion of $Lepr$ in POMC neurons impairs body weight and fat mass, the direct contribution of these neurons in regulating glucose homeostasis has always been hard to dissect. Here, we show that deleting LEPRs from POMC neurons in adult mice does not affect body weight or body composition (*Figure 2A–C*). Four weeks following the deletion, we evaluated food intake and energy expenditure using metabolic cages. We observed that food intake was unchanged in mice lacking LEPRs in adult POMC neurons (*Figure 2D*). Moreover, oxygen consumption, respiratory exchange ratio ($VCO_2/VO_2$) and physical activity were all unaltered (*Figure 2E–G*). These results suggest that LEPR-expressing POMC neurons regulate liver insulin sensitivity independently of changes in body weight.

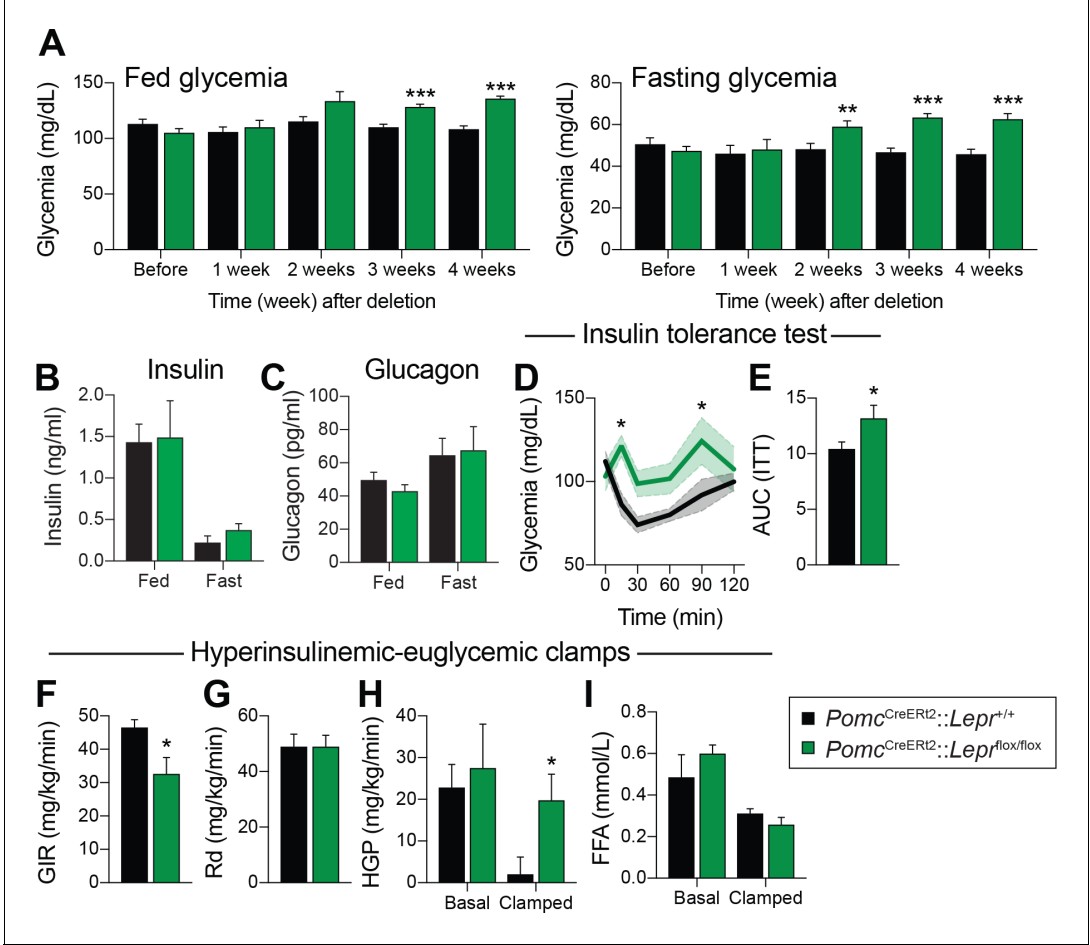

**Figure 1.** LEPR-expressing POMC neurons are required for normal liver insulin sensitivity in adult mice. (**A**) Fed and fasting (16 hr) glucose one week before, and every week for four weeks after, *Pomc*<sup>CreERt2</sup>*::Lepr*<sup>flox/flox</sup> and littermate controls were injected with the last dose of tamoxifen (n = 12). (**B**) Fed and fasting (48 hr) insulin four weeks after tamoxifen was given (n = 4–6). (**C**) Fed and fasting (48 hr) glucagon four weeks after tamoxifen was given (n = 4–6). (**D**) Glucose excursion during an insulin tolerance test (ITT) only one week following the last injection of tamoxifen (n = 5–6). (**E**) Area under the curve for the ITT shown in B (n = 5–6). (**F**) Glucose infusion rate (GIR) needed to maintain euglycemia (119.3 ± 3.9 vs 122.0 ± 8.2 mg/dl) during an hyperinsulinemic-euglycemic clamp performed only one week following the last injection of tamoxifen (n = 6). (**G**) Glucose disposal (Rd) during the same hyperinsulinemic-euglycemic clamp (n = 6). (**H**) Basal and clamped hepatic glucose production (HPG) (n = 6). (**I**) Basal and clamped lipolysis rate as assessed by measuring free fatty acid (FFA) levels (n = 6). The data are expressed as the mean ± SEM. ***p<0.001, **p<0.01 and *p<0.05 versus littermate controls.

DOI: https://doi.org/10.7554/eLife.33710.002

The following figure supplements are available for figure 1:

**Figure supplement 1.** Validation of the *Pomc*<sup>CreERt2</sup> mice.

DOI: https://doi.org/10.7554/eLife.33710.003

**Figure supplement 2.** Glucagon stimulation test.

DOI: https://doi.org/10.7554/eLife.33710.004

Fasting reduces *Pomc* mRNA expression in the ARC (*Mizuno et al., 1998*), and this reduction contributes to the promotion of hunger (*Mercer et al., 2013*). We found that adult deletions of LEPRs in POMC neurons did not affect fed or fasting levels of *Pomc* mRNA (*Figure 3A*). Another population of hypothalamic neurons that regulate energy balance and glucose homeostasis are the orexigenic neuropeptide Y (NPY)/agouti related peptide (AgRP) neurons (*Schwartz et al., 2000*; *Morton et al., 2006*). During fasting, the activity of these neurons increases, which promotes food-seeking and eating behaviors (*Takahashi and Cone, 2005*). Moreover, leptin inhibits NPY/AgRP neurons and fasting relieves this inhibition (*Schwartz et al., 1996*). Interestingly, mice with adult deletions of LEPRs in POMC neurons had blunted mRNA levels of *Npy* and *Agrp* in response to

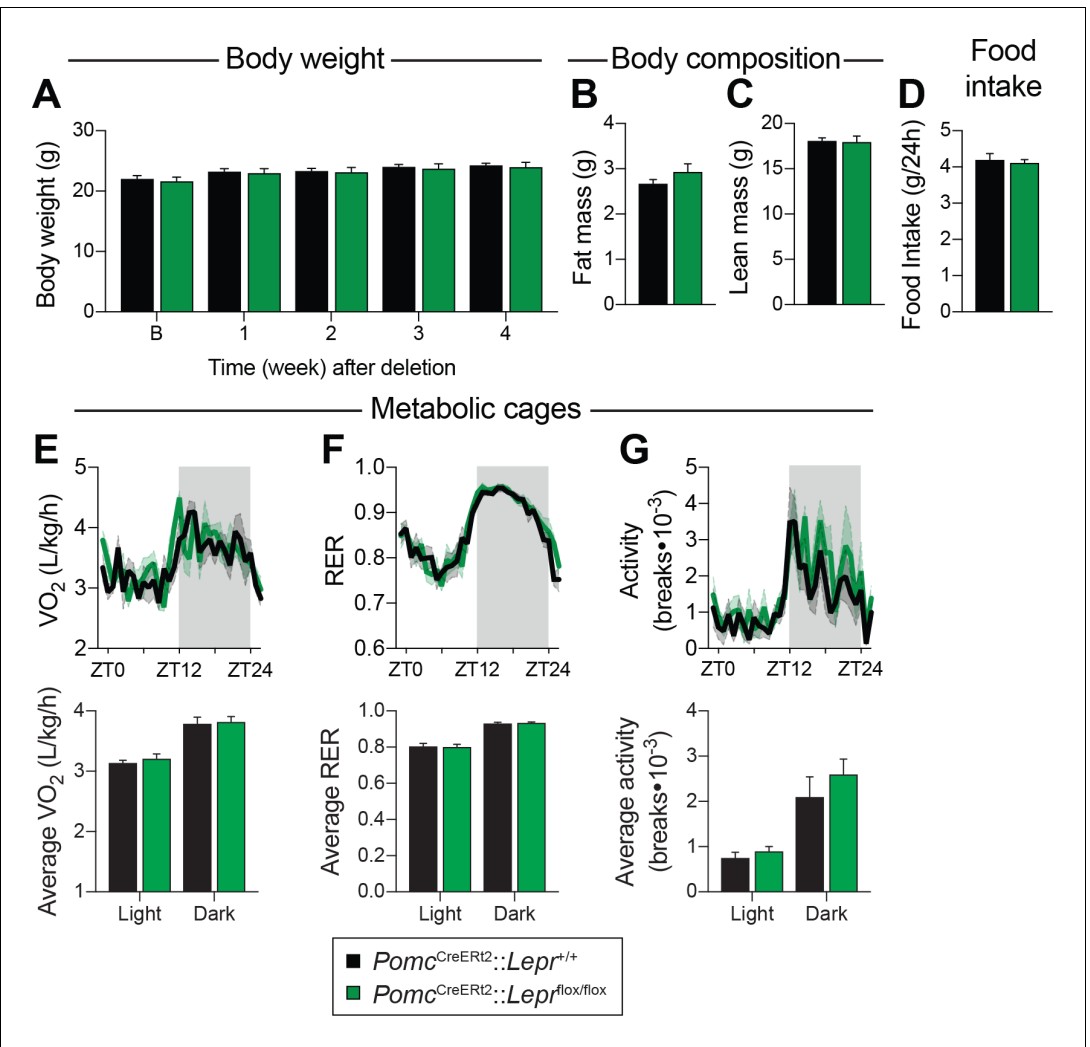

**Figure 2.** LEPR-expressing POMC neurons are dispensable for the regulation of energy balance in adult mice. (A) Body weight before, and up to four weeks after, $Pomc^{CreERt2}::Lepr^{flox/flox}$ and littermate controls were injected with tamoxifen (n = 12). (B) Fat mass and C) Lean mass as assessed by nuclear magnetic resonance (NMR) four weeks following tamoxifen administration (n = 12). (D) Daily food intake, (E) Oxygen consumption ($VO_2$), (F) Respiratory exchange ratio (RER), and G) locomotor activity in CaloSys Calorimetry System cages four weeks after the administration of tamoxifen (n = 5). Summary graphs showing average data for light (ZT0-ZT12) and dark (ZT12-ZD24) cycles are presented under each diurnal graph. The data are expressed as the mean ± SEM.
DOI: https://doi.org/10.7554/eLife.33710.005

starvation (*Figure 3B–C*). This suggests that despite normal food intake in unrestrained conditions (*Figure 2*), fasting-induced hyperphagia might be impaired in mice lacking LEPR in adult POMC neurons. However, we found that mice consumed the same amount of food when access to laboratory chow was restored after a 48 hr fast (*Figure 4A*). Interestingly, feeding-induced hyperglycemia was higher in mice lacking LEPRs in adult POMC neurons (*Figure 4B*). Together, these results reinforce the idea that LEPR-expressing POMC neurons are dispensable for the regulation of energy balance in adult mice. Moreover, these data further demonstrate impaired glucose homeostasis when LEPRs are deleted from adult POMC neurons. At this point, it remains unclear whether manipulating LEPR-expressing POMC neurons results in dysfunction of NPY/AgRP neurons or if the receptors themselves are critical for the fasting response.

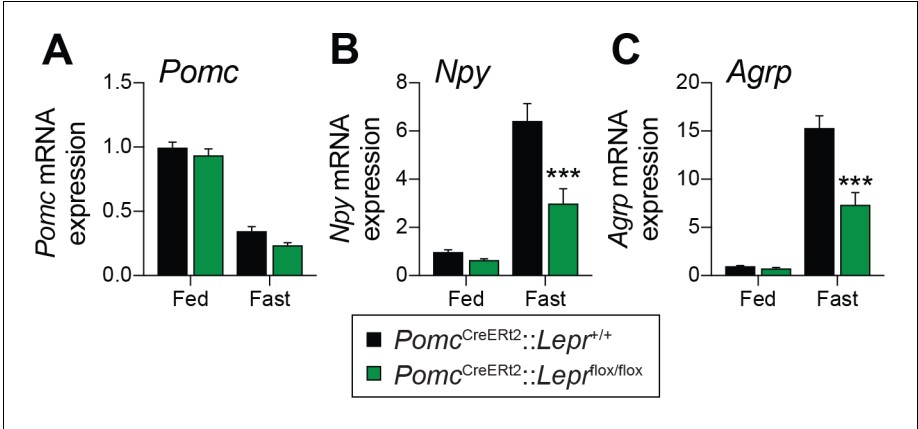

**Figure 3.** Deletion of LEPRs in adult POMC neurons impairs fasting-induced expression of orexigenic neuropeptides in the mediobasal hypothalamus. (A) *Pomc,* (B) *Agrp* and (C) *Npy* mRNA expression in mediobasal hypothalamus of fed and fasted (48 hr) *Pomc*[CreERt2]*::Lepr*[flox/flox] and littermate control mice four weeks after tamoxifen was given (n = 8–14). The data are expressed as the mean ± SEM. ***p<0.001 versus littermate controls.
DOI: https://doi.org/10.7554/eLife.33710.006

## LEPR-expressing POMC neurons are required for the fasting-induced fall in leptin levels independent of changes in fat mass in adult mice

Fasting leads to a rapid fall in circulating leptin levels, despite no initial changes in fat mass (*Becker et al., 1995*; *Moinat et al., 1995*; *Saladin et al., 1995*; *Ahima et al., 1996*; *Flier, 1998*; *Ahima et al., 1999*). However, this regulation in leptin levels is blunted in LEPR-null animals (*Hardie et al., 1996*), suggesting that LEPRs per se are required for the starvation-induced fall in leptin. In order to better understand the potential contribution of LEPR-expressing POMC neurons in regulating leptin production, we compared the impact of 48 hr of fasting in prenatal and adult models. In contrast to prenatal deletions (*Figure 5A–B*), deleting LEPRs in POMC neurons in adult mice did not affect fasting-induced decreases in body weight or fat-mass loss (*Figure 5E–F*). Consistent with previous reports (*Moinat et al., 1995*; *Trayhurn et al., 1995*; *Ahima et al., 1996*;

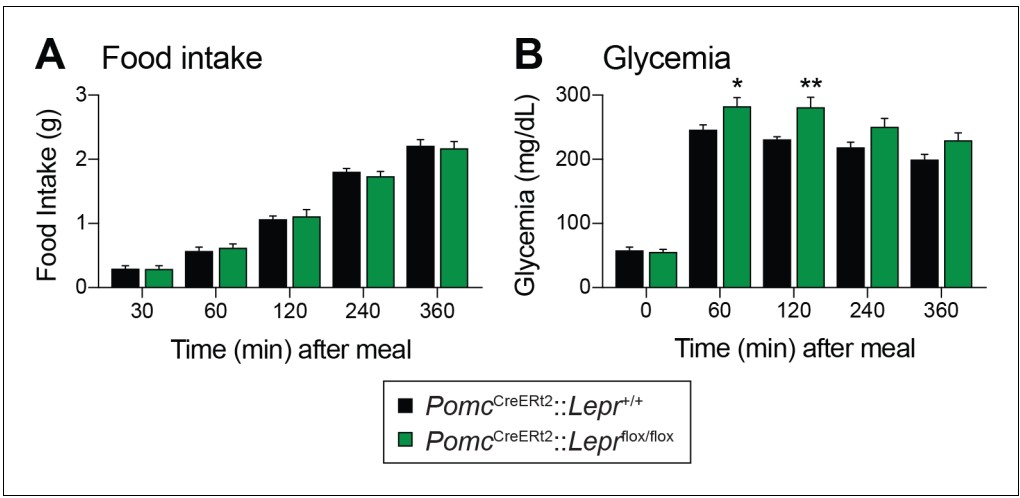

**Figure 4.** Deletion of LEPRs in adult POMC neurons impairs postprandial glycemia. (A) Food intake and (B) Blood glucose up to six hours after food access was restored to 48 hr fasted *Pomc*[CreERt2]*::Lepr*[flox/flox] and littermate control mice, four weeks after tamoxifen was given (n = 8). The data are expressed as the mean ± SEM. **p<0.01 and *p<0.05 versus littermate controls.
DOI: https://doi.org/10.7554/eLife.33710.007

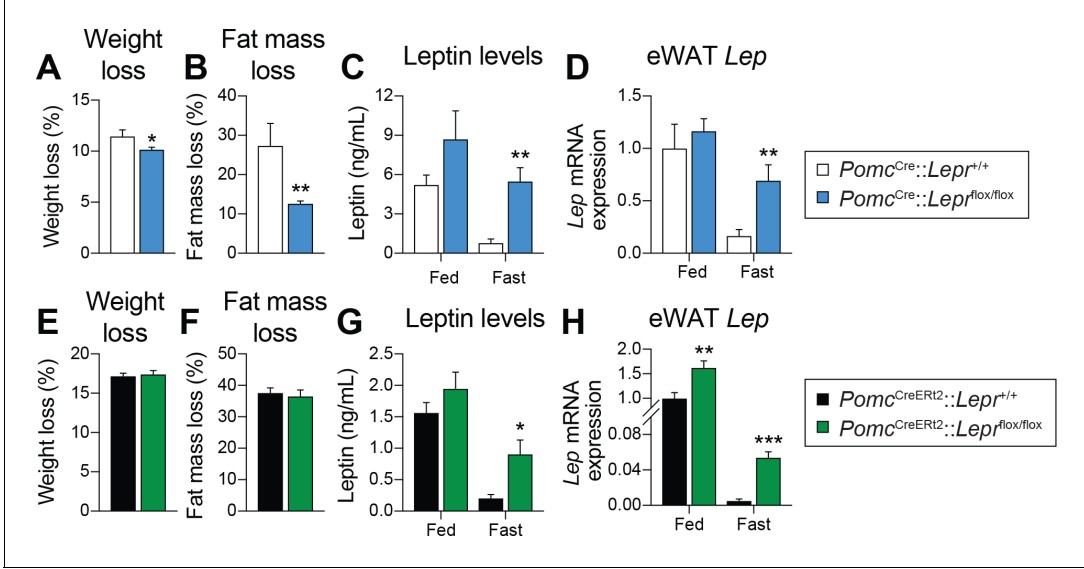

**Figure 5.** LEPR-expressing POMC neurons are required for the fasting-induced fall in leptin levels, independent of changes in fat. (A) Weight loss, and (B) fat-mass loss after a 48 hr fast in mice with constitutive (prenatal) deletion of LEPRs in POMC neurons and littermate controls (n = 7–10). (C) Plasma leptin levels, and D) visceral adipose tissue *Lep* mRNA expression in fed or fasted (48 hr) mice with constitutive deletion of LEPRs in POMC neurons and littermate controls (n = 7–14). (E) Weight loss, and (F) fat-mass loss after a 48 hr fast in *Pomc*^CreERt2^::*Lepr*^flox/flox^ and littermate control mice four weeks after tamoxifen was given (n = 12). (G) Plasma leptin levels, and (H) visceral adipose tissue *Lep* mRNA expression in fed or fasted (48 hr) in *Pomc*^CreERt2^::*Lepr*^flox/flox^ and littermate control mice four weeks after tamoxifen was given (n = 6–13). The data are expressed as the mean ± SEM. ***p<0.001, **p<0.01 and *p<0.05 versus littermate controls.

DOI: https://doi.org/10.7554/eLife.33710.008

*Hardie et al., 1996*), fasting induced a robust fall in both circulating leptin and visceral adipose *Lep* mRNA levels in wild-type littermate controls (*Figure 5C–D and G–H*). Strikingly, this effect was prevented in mice with either prenatal (*Figure 5C–D*) or adult (*Figure 5G–H*) deletions of LEPRs in POMC neurons. Although modest, expression of *Lep* in visceral adipose tissue was significantly higher, in fed mice lacking LEPRs in adult POMC neurons (*Figure 5D and H*), suggesting that the deletion may affect leptin regulation even in the fed state. Collectively, these results indicate that LEPR-expressing POMC neurons are required for the starvation-induced fall in leptin, independent of changes in fat mass. Preventing fasting-induced falls in leptin might explain the blunted response observed in *Agrp* and *Npy* expression (*Figure 3B–C*).

## Gi-coupled alpha-2A adrenergic receptors (ADRA2A) regulate leptin synthesis

Given that adult deletions of LEPRs in POMC neurons are sufficient to prevent the fasting-induced fall in circulating leptin levels, we next sought to determine how these neurons regulate leptin production in adipose tissue. One way the CNS may negatively regulate leptin is through the activation of ADRB3 (*Moinat et al., 1995*; *Collins and Surwit, 1996*; *Gettys et al., 1996*; *Giacobino, 1996*; *Mantzoros et al., 1996*; *Trayhurn et al., 1996*; *Trayhurn et al., 1998*; *Evans et al., 1999*). In addition, overexpression of ADRA2 in mouse adipose tissue increases leptin levels (*Valet et al., 2000*), suggesting that the ADRA2/ADRB3 balance in adipocytes is critical for regulation of leptin. We first investigated the expression of the nine identified adrenergic receptors in visceral adipose tissue (*Figure 6A*). The expression of most of the adrenergic receptors was unchanged in mice with adult deletions of LEPRs in POMC neurons compared to wild-type littermates. However, the fasting-induced decrease in *Adra2a* mRNA expression was not only prevented, but reversed following the deletion of LEPRs in adult POMC neurons (*Figure 6A*). Using an independent cohort, we found that this observation was not only reproducible, but also specific to visceral adipose tissue (*Figure 6B–C*). This result is in line with the fact that visceral but not subcutaneous adipose tissue is the primary

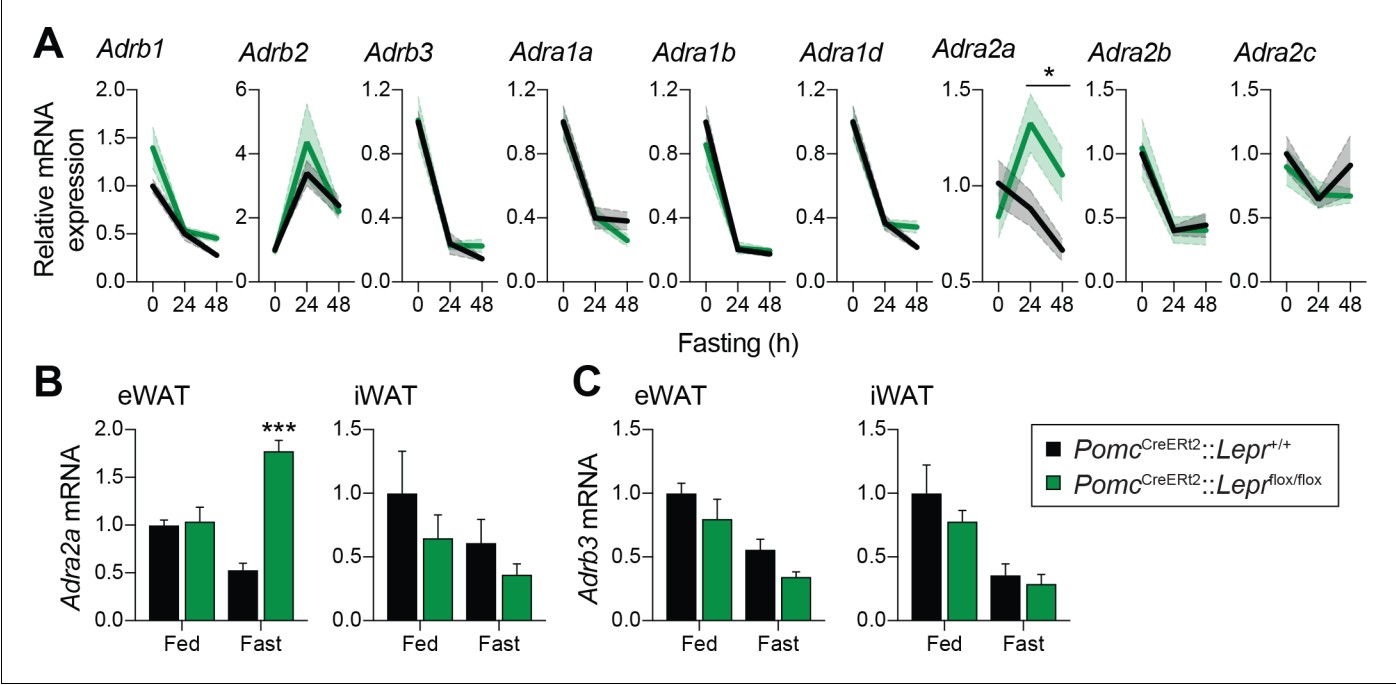

**Figure 6.** Deletion of LEPRs in adult POMC neurons impairs visceral adipose tissue expression of *Adra2a* with fasting. (A) Expression of the nine adrenergic receptors in fed of fasted (24 hr – 48 hr) *Pomc*CreERt2::*Lepr*flox/flox and littermate control mice four weeks after tamoxifen was given (n = 8–14). (B) Comparison of the expression of *Adra2a* and (C) *Adrb3* in epididimal (eWAT) versus inguinal (iWAT) adipose tissue in an independent cohort of fed of fasted (48 hr) *Pomc*CreERt2::*Lepr*flox/flox and littermate control mice four weeks after tamoxifen was given (n = 5–6). The data are expressed as the mean ± SEM. ***p<0.001 and *p<0.05 versus littermate controls.

DOI: https://doi.org/10.7554/eLife.33710.009

source of leptin in rodents (*Trayhurn et al., 1995*). These findings suggest that ADRA2A may be a candidate for mediating the starvation-induced fall in leptin.

The function of ADRA2A in adipocyte physiology and pathophysiology is well known (*Lafontan and Berlan, 1995*; *Garg et al., 2016*). However, its role in leptin synthesis has never been investigated. To functionally validate a role for ADRA2 in regulating leptin expression and production, C57BL/6J mice were intraperitoneally injected with the ADRA2 agonist clonidine, and visceral adipose tissue was collected 1 hr later. Strikingly, clonidine increased *Lep* mRNA expression by six fold (*Figure 7A*). In another cohort of C57BL/6J mice, we also observed that clonidine rapidly increased plasma leptin levels (*Figure 7B*). We next sought to evaluate whether clonidine treatment altered leptin production in mice with adult deletions of LEPRs in POMC neurons. Because clonidine affects every ADRA2, including those express in the CNS, we performed the experiment using adipose tissue explants from mice that were fed or fasted for 48 hr prior to the euthanasia. In fed animals, we found higher leptin release in knock-out animals (*Figure 7C*), consistent with the higher expression of *Lep* mRNA observed in visceral adipose tissue (*Figure 5D,H*). Furthermore, in the fasted condition, clonidine was effective at inducing leptin release only in adipose tissue explants from mice with LEPRs deleted in adult POMC neurons (*Figure 7C*). These explant studies indicate that this effect is adipose tissue-autonomous and not mediated through central effects. These results are in line with the observation that *Adra2a* mRNA expression increases with fasting in visceral adipose tissue of knock-out animals (*Figure 6A*). Clonidine was ineffective in subcutaneous adipose tissue (*Figure 7D*), again suggesting that the regulation of leptin production is specific to visceral fat. Together, these results suggest a role for ADRA2 as critical regulator of both leptin expression and production. In addition, these data suggest that ablation of LEPRs in adult POMC neurons prevents the starvation-induced fall in leptin by increasing ADRA2A activity in visceral white adipose tissue.

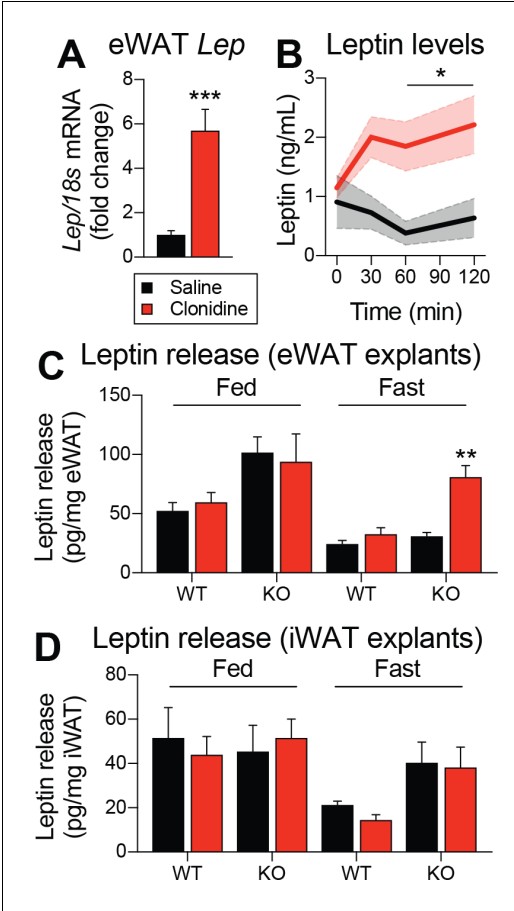

**Figure 7.** Pharmacological activation of ADRA2 stimulates leptin production. (**A**) Visceral adipose tissue *Lep* mRNA expression one hour following an intraperitoneal (1 mg/kg) injection of the ADRA2 agonist clonidine (n = 10–12). (**B**) Plasma leptin levels up to two hours following the administration of clonidine in an independent cohort (n = 4–7). (**C**) Leptin release from epididymal (eWAT) and (**D**) inguinal (iWAT) adipose tissue explants from fed and fasted (48 hr) *Pomc*^CreERt2::*Lepr*^flox/flox and littermate control mice following the addition of clonidine (1 μM) (n = 6). This experiment was performed four weeks after tamoxifen was given. The data are expressed as the mean ± SEM. ***p<0.001, **p<0.01, and *p<0.05 versus littermate controls.

DOI: https://doi.org/10.7554/eLife.33710.010

## Discussion

Leptin signaling in POMC neurons has been predicted to be key in regulating energy balance and glucose homeostasis (*Münzberg et al., 2003*; *Balthasar et al., 2004*; *Kievit et al., 2006*; *Huo et al., 2009*; *Berglund et al., 2012*; *Huang et al., 2012*; *Mercer et al., 2013*). Our current findings dissociate the effects of LEPR-expressing POMC neurons on glucose homeostasis and changes in energy balance. In addition, our results suggest that POMC neurons are key regulators of leptin levels. This is interesting as one of the questions in leptin biology is the mechanism behind starvation-induced falls in leptin (*Friedman, 2016*; *Beshel et al., 2017*). Although it may appear paradoxical that a subset of LEPR-expressing POMC cells controls leptin synthesis, previous studies have suggested that LEPRs are required for the starvation-induced fall in leptin (*MacDougald et al., 1995*; *Hardie et al., 1996*; *Commins et al., 2000*). This supports previous models that falling leptin is required to activate neuroendocrine responses (*Ahima et al., 1996*; *Ahima et al., 1999*). We also identify a role for ADRA2A in regulating leptin levels during starvation. This is in agreement with a report in which expression of human ADRA2A in adipocytes resulted in elevated leptin levels (*Valet et al., 2000*). Collectively, our study highlights a previously unrecognized role of POMC neurons in the regulation of leptin levels and provides a new framework for the understanding of leptin action and regulation in the context of changing states of energy balance.

The current study highlights the ongoing importance of developing more refined transgenic tools, including adult-inducible models. Here, we used a tamoxifen-inducible *Pomc*^CreERt2 transgenic mouse model to generate mice in which *Lepr* expression is spatiotemporally controlled in a neuron-specific fashion. Recent findings have demonstrated a need for the development of such a tool. First, the central melanocortin pathways are developmentally plastic, and as such compensations might affect the resulting phenotype, inherently limiting the conclusions that can be drawn (*Bouret et al., 2004*; *Wu et al., 2009*; *Padilla et al., 2010*; *Bouret et al., 2012*; *Wu et al., 2012*). In addition, POMC neurons share developmental origin with other cell types, including their NPY/AgRP counterparts (*Padilla et al., 2010*). For instance, over 25% of POMC-positive neurons were shown to express high levels of *Agrp* (*Lam et al., 2017*). Likewise, we recently developed an *Agrp*^CreERt2 transgenic mouse model to better study the role of AgRP neurons in ghrelin response (*Wang et al., 2014*). These inducible tools will allow us to revisit fundamental beliefs about the central melanocortin system.

The canonical effect of leptin action in the brain is to regulate energy balance (*Millington, 2007*; *Mercer et al., 2013*). Despite early evidence that ablating LEPRs only in POMC neurons results in

moderate changes in body weight (*Balthasar et al., 2004*), leptin action on POMC neurons in the ARC is considered a prototypical site of action in the control of food intake and energy expenditure. We and others have previously proposed that leptin directly acts on POMC neurons to regulate glucose homeostasis (*Huo et al., 2009*; *Berglund et al., 2012*). There is also evidence that subpopulations of POMC neurons that do not express LEPRs may regulate food intake (*Xu et al., 2008*; *Williams et al., 2010*; *Berglund et al., 2013*; *Campbell et al., 2017*). It is also possible that the mild obesity observed in previous studies is the consequence of *Lepr* deletion from a proportion of AgRP neurons. Our data indicate that the effects of leptin on energy balance are not through direct actions on POMC neurons.

Here we show that action of leptin on POMC neurons regulates glucose homeostasis independent of its effects on energy balance. Specifically, removing LEPRs from POMC neurons in adult mice resulted in insulin resistance and impaired hepatic glucose production within one week following deletion. This was followed by sustained hyperglycemia, independent of changes in insulin and glucagon levels, in glucose disposal, or in the ability of insulin to suppress lipolysis. Although food intake was unaltered both in *ad libitum* or refeeding conditions, postprandial glycemia was impaired in mice lacking LEPRs in adult POMC neurons. Together, this suggests that altering leptin signaling in POMC neurons results in rapid-onset hepatic insulin resistance (*Brown and Goldstein, 2008*). This specific effect is consistent with many reports showing direct consequences in the liver following genetic manipulations in POMC neurons (*Hill et al., 2010*; *Xu et al., 2010*; *Berglund et al., 2012*; *Berglund et al., 2013*; *Shi et al., 2013*; *Williams et al., 2014*; *Caron et al., 2016*). It was also recently shown that POMC neurons are important for hepatic parasympathic nerve activity in response to leptin (*Bell et al., 2018*). A recent study also stresses the importance of insulin signaling in POMC neurons in regulating adipose tissue lipolysis and the development of liver steatosis (*Shin et al., 2017*). However, whether POMC neurons regulate glucose and lipid hepatic metabolism directly through the autonomic nervous system, or indirectly by altering metabolic hormone requires further investigation. It is nevertheless clear from our study that LEPR-expressing POMC neurons play a pivotal role in liver metabolism, independently of changes in energy balance.

Our data also highlight an unexpected role for LEPR-expressing POMC neurons in regulating the fasting-induced fall in leptin. We show that the ability of fasting to suppress leptin is impaired in transgenic mouse models with either prenatal or adult deletion of LEPRs in POMC neurons. Although there is a general consensus that leptin levels are tightly correlated to adiposity (*Frederich et al., 1995*; *Considine et al., 1996*), our data suggest that this fasting-dependent regulation is independent of changes in body weight or fat mass. Moreover, this effect appears specific to visceral adipose tissue, which is in line with the fact that leptin is predominantly secreted from visceral white adipocytes in rodents (*Trayhurn et al., 1995*).

However, one important question still remains. In a particular, how does LEPR signalling in POMC neurons regulate adipocyte leptin secretion during fasting? One speculation is that the deletion of leptin receptors reduces POMC activity and renders the neurons less effective at activating downstream targets. Another possibility is more provocative. In particular, we propose that LEPR-expressing POMC neurons are part of a regulatory loop that is important for adaptative responses to fasting. Fasting rapidly alters key metabolic signals and decreases the circulating peripheral hormones (such as insulin) which are required to maintain normal leptin levels (*Saladin et al., 1995*; *D'souza et al., 2017*). These changes are all sensed by POMC neurons. However, drops in leptin trigger neuroendocrine responses that promote survival, including the inhibition of the sexual and thyroid axes and activation of the stress axis (*Ahima et al., 1996*; *Ahima et al., 1999*). These survival responses are extreme and safeguards may have evolved to ensure that they are not initiated too quickly. LEPR-expressing POMC neurons might represent such a 'gatekeeper' to control the inhibition of leptin production. Thus, removing LEPRs from POMC neurons would prevent their ability to sense small fluctuations in leptin levels ultimately blunting the ability to fully suppress leptin levels.

In support of this model, we observed that fasting-induced expression of *Npy* and *Agrp* in the mediobasal hypothalamus was impaired in *Pomc*^CreERt2^::*Lepr*^flox/flox^ mice, suggesting that the falling leptin might be an important signal activating NPY/AgRP neurons during starvation (*Bi et al., 2003*). Although this impaired response might be a direct consequence of elevated fasting-leptin levels, we did not observe any differences in food intake. Importantly, this does not invalidate the role of these neurons in regulating re-feeding behavior after a fast. However, these results indicate that

preventing the normal fall in leptin levels during fasting have major repercussions, not only on the neuroendocrine system (*Ahima et al., 1996*), but also on behavioral, metabolic and neuronal responses.

Mechanistically, we show that visceral adipose tissue expression of *Adra2a*, which normally decreases with fasting, is actually increasing in mice lacking fasted mice lacking LEPRs in adult POMC neurons. Interestingly, the expression of *Adra2a* is not altered in subcutaneuous adipose tissue, further supporting visceral-dependent effect. It is noteworthy that the sympathetic regulation differs between different fat depots, both in terms of innervation and outflow (*Brito et al., 2007*; *Brito et al., 2008*; *Nguyen et al., 2017*). These findings also add another layer of complexity to the way the brain regulates peripheral tissues through the activation of GPCRs. Our pharmacological experiments also support the notion that ADRA2 are important for leptin regulation. ADRB3 is well-known to negatively regulate leptin though a cAMP-dependent mechanism (*Moinat et al., 1995*; *Gettys et al., 1996*; *Giacobino, 1996*; *Mantzoros et al., 1996*; *Slieker et al., 1996*; *Trayhurn et al., 1996*; *Deng et al., 1997*; *Trayhurn et al., 1998*; *Caron et al., 2018*). Because ADRB3 is Gs-coupled, we hypothesize that Gi-coupled ADRA2 might have the opposite action on leptin synthesis. Treating mice with an ADRA2 agonist is sufficient to increase both circulating leptin and mRNA levels in visceral fat. We also found that this regulation is tissue-autonomous, as clonidine effectively affected leptin release only in visceral adipose tissue explants from mice lacking LEPRs in adult POMC neurons. From a translational point of view, the observation that ADRA2A activation stimulates leptin production is meaningful. Human adipocytes express high levels of ADRA2A but few or no ADRB3, while murine adipocytes show high levels of ADRB3 and very low number of ADRA2 (*Lafontan and Berlan, 1993*; *Lafontan and Berlan, 1995*). By creating mice that have a human-like pattern of adrenoreceptors, researchers previously established that the ADRA2/ADRB2 balance in adipocytes is critical for regulating fat mass (*Valet et al., 2000*). Increasing the ADRA2/ADRB3 balance in adipose tissue resulted in increased circulating levels of leptin, suggesting that this balance is also important for regulating leptin production. However, because these mice were obese, the direct contribution of the ADRA2/ADRB3 balance was hard to define. Here, we show that despite no changes in body weight, the ADRA2/ADRB3 balance in adipocyte is still important for leptin regulation.

In conclusion, our study indicates that a subset POMC neurons that express LEPRs directly controls glucose homeostasis and is necessary to control leptin synthesis, independently of changes in fat mass. We also identified an important role for adipose tissue ADRA2A in regulating leptin synthesis. From a conceptual standpoint, our results predict that leptin regulates its own expression through a negative feedback loop between POMC neurons and adipose tissue.

# Materials and methods

**Key resources table**

| Reagent type (species) or resource | Designation | Source or reference | Identifiers |
|---|---|---|---|
| Strain (Tg(Pomc-cre)1Lowl) | *Pomc*Cre mouse | PMID: 17556551 | RRID:IMSR_JAX:010714 |
| Strain (Tg(Pomc-cre/ERT2)#Jke) | *Pomc*CreERt2 mouse | PMID: 24177424 | RRID:MGI:5569339 |
| Strain (Leprtm1.1Chua) | *Lepr*flox/flox mouse | PMID: 15389315 | RRID:MGI:3511747 |
| Strain (Gt(ROSA)26Sortm14 (CAG-tdTomato)Hze) | Ai14(RCL-tdT)-D mouse | PMID: 20023653 | RRID:IMSR_JAX:007914 |
| Antibody (AB_331586) | phospho-Stat3 antibody | Tyr705, Cell Signaling Technology Cat# 9131, | RRID:AB_331586 |
| Antibody (AB_2314007) | β-endorphin antibody | Phoenix Pharmaceuticals Cat# H-022–33 | RRID:AB_2314007 |
| Antibody (AB_639922) | tdTomato antibody | Santa Cruz Biotechnology Cat# sc-33354, | RRID:AB_639922 |

## Animals

Animal work described in this manuscript has been approved and conducted under the oversight of the UT Southwestern Institutional Animal Care and Use Committee (IACUC). Male mice were housed

at an ambient temperature of 23 ± 1°C and maintained on a 12 hr light/dark cycle (lights on 0700–1900) and fed with normal mouse chow diet (Harlan, Teklad Global 16% Protein Rodent Diet 2016; 12% kcal from fat, 3 kcal/g).

$Pomc^{Cre}$ (RRID:IMSR_JAX:005965) mice (*Balthasar et al., 2004*) and $Pomc^{CreERt2}$ (RRID: MGI:5569339) mice (*Berglund et al., 2013*) were crossed with $Lepr^{flox/flox}$ (RRID:MGI:3511747) mice (*McMinn et al., 2004*) to generate mice with constitutive deletion of LEPRs in POMC neurons ($Pomc^{Cre}$::$Lepr^{flox/flox}$) and adult deletion of LEPRs in POMC neurons ($Pomc^{CreERt2}$::$Lepr^{flox/flox}$) respectively. Mice were maintained on a C57Bl/6J (RRID:IMSR_JAX:000664) background at UT Southwestern Medical Center. Adult ablation was induced by tamoxifen. Tamoxifen (0.15 mg/g; Sigma-Aldrich, T5648) was suspended in corn oil (Sigma-Aldrich, C8267) and was administered intraperitoneally (three injections every 48 hr for 5 days) to 10–12 week-old $Pomc^{CreERt2}$::$Lepr^{flox/flox}$ and $Pomc^{CreERt2}$::$Lepr^{+/+}$ (littermate control) mice. Fasting experiments were performed from 0800 to 0800 (48 h) or from 1600 to 0800 (16 h). The efficiency of the recombination following tamoxifen was performed by crossing $Pomc^{CreERt2}$ mice with Ai14(RCL-tdT)-D mice (RRID:IMSR_JAX:007914) mice. Validation of the mouse model is presented in *Figure 1—figure supplement 1*.

## Immunohistochemistry and validation of the inducible mice

Immunohistochemistry was performed to visualize phospho-Stat3 (Tyr705, Cell Signaling Technology Cat# 9131, RRID:AB_331586), β-endorphin (Phoenix Pharmaceuticals Cat# H-022–33, RRID:AB_2314007), as well as the fluorescent reporter tdTomato (Santa Cruz Biotechnology Cat# sc-33354, RRID:AB_639922) in the brain and pituitary (*Scott et al., 2009*; *Williams et al., 2010*). For leptin-induced Stat3 activation experiments, mice were fasted for 16 hr (1600 to 0800) and injected i.p. with mouse recombinant leptin (5 mg/kg; National Hormone and Peptide Program, AFP1783). Mice were anesthetized 45 min later using an i.p. injection of chloral hydrate (350 mg/kg) and then perfused transcardially with 0.9% saline followed by 10% neutral buffered formalin.

## Assessment of insulin sensitivity and glucose levels

Blood samples were collected from the tail vein and glucose was measured using a glucometer (Bayer's Contour Blood Glucose Monitoring System; Leverkusen, Germany). For insulin tolerance test (ITT), mice were fasted for 4 hr and then administered insulin by intraperitoneal injection (0.75 U/kg body weight, human insulin, Eli Lilly).

## Hyperinsulinemic-euglycemic clamps

Hyperinsulinemic-euglycemic clamps were performed on conscious, unrestrained mice as previously described (*Holland et al., 2011*). Euglycemia was maintained by variable infusion of 20% dextrose. Steady state was achieved 80 min after initiating hyperinsulinemia and maintained for 40 min. Additional blood samples were taken before initiating hyperinsulinemia and at the end of the clamp for analysis of insulin and free fatty acids.

## Glucagon stimulation test

Glucagon stimulation test was performed in mice fasted for one hour (0800 to 0900). Briefly, human recombinant glucagon (120 µg/kg i.p.) was given and blood glucose monitored every 10 min for one hour.

## Assessment of leptin, insulin and glucagon levels

Blood was collected in EDTA tubes. Plasma was isolated by centrifugation (4000 g x 10 min at 4°C) and was stored at −80°C for further biochemical analyses. Plasma leptin (Mouse/Rat Leptin ELISA, ALPCO, 22-LEPMS-E01), insulin (Mouse Ultrasensitive Insulin ELISA, ALPCO, 80-INSMSU-E01), and glucagon (Mercodia Glucagon ELISA, 10-1281-01) were measured following manufacturer recommendations.

## Assessment of body composition

Fat mass and lean mass were assessed by nuclear magnetic resonance (NMR) spectroscopy using a nuclear magnetic resonance (NMR) spectroscopy (Bruker Minispec mq10 NMR 0.23T/10MHz).

## Metabolic cages studies

A combined indirect calorimetry system (CaloSys Calorimetry System, TSE Systems Inc.) was used for all metabolic studies. Experimental animals were acclimated for 5 days in a metabolic chamber with food and water. Oxygen consumption (VO2), carbon dioxide production (VCO2), respiratory exchange ration (RER) and food intake were measured after acclimation. Locomotion was measured using a multi-dimensional infrared light beam system.

## Quantitative real-time PCR

Total mRNA was isolated from visceral (epidymal) and subcutaneous (inguinal) white adipose tissues using the RNeasy Lipid Tissue Mini Kit (Qiagen, 74104). Total mRNA was isolated from liver using RNA STAT-60 reagent (Tel-Test, Inc). The RNA concentrations were estimated from absorbance at 260 nm. cDNA synthesis was performed using a High Capacity cDNA Kit (Applied Biosystems). mRNA extraction and cDNA synthesis were performed following the manufacturer's instructions. cDNA was diluted in DNase-free water before quantification by real-time PCR. mRNA transcript levels were measured in duplicate samples using a ABI 7900 HT Sequence Detection System (Applied Biosystems). The relative amounts of all mRNAs were calculated using the ΔΔCT assay. Primers for *18* s (Hs99999901_s1), *Adra1a* (Mm00442668_m1), *Adra1b* (Mm00431685_m1), *Adra1d* (Mm01328600_m1), *Adra2a* (Mm00845383_s1), Adra2b (Mm00477390_s1), *Adra2c* (Mm00431686_s1), *Adrb1* (Mm00431701_s1), Adrb2 (Mm02524224_s1), *Adrb3* (Mm02601819_g1), *Agrp* (Mm00475829_g1), *Lep* (Mm00434759_m1), *Npy* (Mm00445771_m1) and *Pomc* (Mm00435874_m1) were purchased from Applied Biosystems.

## Pharmacological activation of ADRA2 in vivo

The ADRA2 agonist clonidine hydrochloride (Sigma-Aldrich, St. Louis, MO, US; C7897) was administered intraperitoneally (1 mg/kg) to 10–12 week-old C57BL/6J mice following 4 hr of fasting. Two independent cohorts were used to evaluate *Lep* RNA expression and circulating leptin levels.

## Ex vivo leptin release assay

*Pomc*^CreERt2::*Lepr*^flox/flox and *Pomc*^CreERt2::*Lepr*^+/+ (littermate control) mice were fasted for 48 hr and ~10–20 mg of visceral (epidydimal) and subcutaneous (inguinal) white adipose tissues were cultured in 96 wells plate containing 0.200 ml of Krebs-Ringer Bicarbonate Buffer containing 5 mM glucose and 4% fatty acid-free BSA, as described (*Caviglia et al., 2011*). Tissues were subsequently treated either with or without 1 µM clonidine hydrochloride (Sigma-Aldrich, St. Louis, MO, US; C7897) for basal and clonidine conditions respectively, and leptin release was measured by ELISA and corrected to tissue weight.

## Statistical analysis

Data are expressed as mean ± SEM. Comparison between two experimental conditions were analyzed by Student's unpaired t test. Two-way ANOVA followed by Bonferroni post hoc test was used to compare more than two experimental conditions. All statistical tests were performed using GraphPad Prism (version 7.0), and $p < 0.05$ was considered statistically significant.

## Acknowledgements

We thank the Mouse Metabolic Phenotyping Core at UT Southwestern Medical Center at Dallas. This work was supported by the NIH (R37DK053301 to JKE, R01DK114036 to CL, K01DK11164401 to CMC) and by the American Heart Association (14SDG17950008 to TF, 16SDG27260001 to CL). AC is a Canadian Diabetes Association fellow.

## Additional information

#### Competing interests

Joel K Elmquist: Reviewing editor, *eLife*. The other authors declare that no competing interests exist.

## Funding

| Funder | Grant reference number | Author |
| --- | --- | --- |
| National Institute of Diabetes and Digestive and Kidney Diseases | R37DK053301 | Joel K Elmquist |
| Canadian Diabetes Association | NOD_PF-3-15-4756-AC | Alexandre Caron |
| American Heart Association | 14SDG17950008 | Teppei Fujikawa |
| National Institute of Diabetes and Digestive and Kidney Diseases | K01DK11164401 | Carlos M Castorena |
| National Institute of Diabetes and Digestive and Kidney Diseases | R01DK114036 | Chen Liu |
| American Heart Association | 16SDG27260001 | Chen Liu |

The funders had no role in study design, data collection and interpretation, or the decision to submit the work for publication.

## Author contributions

Alexandre Caron, Conceptualization, Data curation, Formal analysis, Investigation, Methodology, Writing—original draft, Writing—review and editing; Heather M Dungan Lemko, Formal analysis, Methodology; Carlos M Castorena, Teppei Fujikawa, Caleb C Lord, Newaz Ahmed, Charlotte E Lee, William L Holland, Investigation, Methodology; Syann Lee, Supervision, Writing—review and editing; Chen Liu, Validation, Investigation, Methodology; Joel K Elmquist, Resources, Supervision, Funding acquisition, Writing—review and editing

## Author ORCIDs

Alexandre Caron (iD) http://orcid.org/0000-0001-6939-6136
Joel K Elmquist (iD) http://orcid.org/0000-0001-6929-6370

## Ethics

Animal experimentation: Animal work described in this manuscript has been approved and conducted under the oversight of the UT Southwestern Institutional Animal Care and Use Committee (IACUC, APN 2015-101301 and APN 2015-101263).

## Decision letter and Author response

Decision letter https://doi.org/10.7554/eLife.33710.013
Author response https://doi.org/10.7554/eLife.33710.014

# Additional files

## Supplementary files

• Transparent reporting form
DOI: https://doi.org/10.7554/eLife.33710.011

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
