## [Decision Letter]

Thank you for submitting your article "POMC neurons expressing leptin receptors coordinate metabolic responses to fasting via suppression of leptin levels" for consideration by *eLife*. Your article has been favorably evaluated by a Senior Editor and four reviewers, one of whom, Richard D Palmiter (Reviewer #5), is a member of our Board of Reviewing Editors. The following individual involved in review of your submission has agreed to reveal their identity: Martin Myers (Reviewer #3).

The reviewers have discussed the reviews with one another and the Reviewing Editor has drafted this decision to help you prepare a revised submission.

The authors of this paper describe important experiments showing that inactivation of POMC neurons in adult mice primarily affects glucose metabolism rather than appetite and body weight. They provide compelling evidence supporting their conclusions. Nevertheless, the four reviewers agree that a few additional details need to be addressed.

Major issues:

1) Authors should quantify how efficiently Lepr is inactivated in POMC neurons and demonstrate that it does not affect other ARC neurons including the AgRP/NPY neurons, because that is the crux of their argument for why conditional gene inactivation in the adult gives different results than during development.

2) The authors should explain how deficient leptin input to Pomc neurons blocks the effect of fasting on leptin secretion.

3) Authors should determine whether the effect of fasting to lower insulin levels is also defective in mutant mice; if so, this could potentially contribute to the defective plasma leptin response.

4) Since leptin action in the brain reverses hyperglucagonemia in rodent models of uncontrolled diabetes, it is possible that deficient leptin action on Pomc cells raises plasma glucagon levels, which in turn increases hepatic glucose production. Measuring glucagon levels would help resolve this possibility.

*Reviewer #2:*

Caron et al., describe the generation and characterization of mice with tamoxifene-inducible deletion of Lepr-expression from POMC-neurons. The animals present with impaired systemic insulin sensitivity caused by a predominat failure of insulin to suppress hepatic glucose production. On the other hand, body weight and energy homeostasis remain unaltered, in contrast to mice with lack of Lepr-expression on POMC-neurons throughout development. Moreover, this study reveals a novel role for leptin action in POMC-neurons in the regulation of fasting-induced decreases of leptin release through altered ADRA2A-signaling.

The study is well performed, and the manuscript clearly describes the results. The conclusions of the authors are firmly supported by the data provided.

Given that POMC-Cre expression throughout development also leads to deletion of a target gene in a proportion of cells, that will later turn into functionally antagonistic AgRP neurons has limited the interpretability of results obtained from mice with cell-type specific Lepr-deletion using this Cre-mouse. Therefore, the results presented here are of clear interest to the field allowing for the distinction of leptin effects mediated exclusively by POMC-neurons. Moreover, the experiments in addition reveal a novel role for leptin action in POMC-neurons to regulate leptin secretion during fasting, independent from changes in fat mass.

Nevertheless, to further improve the manuscript, the authors should include experiments to validate the efficiency and specificity of Lepr-deletion in POMC-neurons in their current model, since this will be of critical importance to interpret the otherwise very interesting set of results.

*Reviewer #3:*

This is an interesting and thorough manuscript in which the authors have used a conditional Pomc-cre to delete Lepr from Pomc cells in adult animals. This has several advantages relative to previous work in which a constitutive cre might be subject to developmental compensation and to the more promiscuous Pomc-cre expression during development. They show that this maneuver does not alter body weight, but impairs the control of blood glucose (presumably via the SNS); this maneuver also alters the central control of adipose tissue leptin production by altering the expression of adrenergic receptors on adipose tissue.

Overall, the results are clear and convincing.

*Reviewer #4:*

This manuscript reports on the phenotype of mice with Pomc-specific deletion of the leptin receptor in adulthood (which averts developmental consequences of germline deletion strategies). In addition to modest effects on glucose homeostasis consistent with previous investigation into Pomc cell function, the principal finding is that Pomc-specific leptin receptor deletion prevents fasting-induced suppression of plasma leptin levels. The paper is well-written and while the data of interest, they are somewhat narrowly focused. The following concerns are identified.

1) The authors do not offer a cogent, physiologically-viable explanation for how it is possible for deficient leptin input to Pomc neurons to block the effect of fasting on leptin secretion. The problem is that fasting rapidly and potently lowers plasma leptin levels, presumably reducing leptin action on Pomc cells as a result. Thus, fasting and the mutant mice are each characterized by reduced leptin action in Pomc cells, and yet they have opposite effects on plasma leptin levels. Some effort to bring clarity to this paradox is needed.

2) The glucose metabolic phenotype is 1) rather modest, 2) confirmatory of previous literature on Pomc dysfunction, and 3) incomplete in the sense that measures of basal, glucose-stimulated and fasting insulin levels were not included. One wonders, for example, whether the effect of fasting to lower insulin levels is also defective in mutant mice; if so, this could potentially contribute to the defective plasma leptin response.

3) While the clamp data support the conclusion that mutant mice have liver insulin resistance, in the sense that insulin did not inhibit HGP normally, such an effect could potentially involve mechanisms independent of insulin action per se. For example, increased SNS tone to liver and/or islet could increase hepatocyte PKA signaling, which in turn would activate both glycogenolysis and gluconeogenesis. This could occur via either direct (e.g., catecholamine release within the liver) or indirect mechanisms (e.g., increased glucagon secretion). Since leptin action in the brain reverses hyperglucagonemia in rodent models of uncontrolled diabetes (Meek, et al., Endocrinology 2013 Sep;154(9):3067-76), it is possible deficient leptin action on Pomc cells raises plasma glucagon levels, which in turn increases HGP.

4) While the data support the conclusion that leptin receptor signaling in Pomc neurons is required for the effect of fasting to lower leptin levels, they do not exclude a comparable role for leptin signaling in other neurons. Since the authors are likely to have other neuron-specific leptin receptor mutants available, it would be useful to survey them to determine whether the observed phenotype is specific to leptin receptor deletion in Pomc cells.

5) Validation of the mutant phenotype is needed. At a minimum, the authors should test whether following tamoxifen, Pomc neurons no longer respond to leptin in terms of pSTAT3 and/or c-Fos induction.

*Reviewer #5:*

This paper describes the effect of inactivating the leptin receptor selectively in POMC neurons in adult mice (rather than during development). Their experiments reveal that Lepr signaling in POMC neurons has no effect on feeding or body weight, but has significant effects on glucose metabolism. Furthermore, the authors show that leptin receptor signaling in POMC neurons affects leptin production by white adipose tissue during fasting by a process that involves α and β adrenergic receptors. These are important observations worthy of publication.

1) The authors show (Figure 3) that Npy and AgRP mRNA levels are significantly reduced after removing LEPR from POMC neurons, but feeding is normal. What is the interpretation of these data? Do the results mean that NPY and AgRP do not regulate feeding behavior after a fast?

2) The extent of Lepr gene inactivation in POMC neurons by tamoxifen treatment should be measured (e.g. loss of leptin-induced pStat3). Since Npy and Agrp mRNA levels are altered, it is also important to show that loss of Lepr is restricted to POMC neurons and not the AgRP/NPY neurons.

---

## [Author Response]

Major issues:1) Authors should quantify how efficiently Lepr is inactivated in POMC neurons and demonstrate that it does not affect other ARC neurons including the AgRP/NPY neurons, because that is the crux of their argument for why conditional gene inactivation in the adult gives different results than during development.

This is an excellent suggestion. We now provide additional data regarding the efficiency of the Pomc-CreERT2 mouse model. First, *Pomc*^CreERT2^ mice were crossed with Ai14(RCL-tdT)-D mice (JAX: 007914). Offspring were injected with tamoxifen, using the described paradigm, to induce the expression of tdTomato in POMC cells. Immunohistochemistry for tdTomato (red) and β-endorphin (a marker of POMC neurons) expression (green) was performed 2 weeks after tamoxifen injection. We show that 311/335 (~96%) β-endorphin+ cells were also TdTomato+, indicating that our approach efficiently targets most POMC neurons (Figure 1—figure supplement 1). In addition, from 350 counted neurons, only 15 (4%) were TdTomato+ only, indicating that our approach does not target a large proportion of non-POMC cells. We also show that our Pomc-CreERT2 model does not induce expression of POMC outside of the hypothalamus, with the exception of a few β-endorphin+ cells located in the nucleus tractus solitarius, a region known to shelter POMC neurons. (Figure 1—figure supplement 1) We also performed a leptin-induced Stat3 activation experiment using Pomc-CreERT2::Lepr^flox/flox^ mice and their littermate controls. Briefly, mice were injected with tamoxifen as described, and leptin (5 mg/kg, i.p.) was administrated in 16h-fasted animals. Animals were decapitated 45 minutes later and pStat3 was evaluated by immunohistochemistry. We show that leptin-induced pStat3 in the ARC area adjacent to the ventricle (where AgRP/NPY neurons mostly reside) is not altered, but that there is a reduction in the most lateral part (where POMC neurons reside) (Figure 1—figure supplement 1). These findings indicate that our approach is unlikely to impair the ability of leptin to signal in AgRP/ NPY neurons. We hope the reviewers will appreciate the validation data we now provide.

2) The authors should explain how deficient leptin input to Pomc neurons blocks the effect of fasting on leptin secretion.

We extended the Discussion on the potential mechanisms by which POMC neurons impairs the ability of fasting to suppress leptin.

3) Authors should determine whether the effect of fasting to lower insulin levels is also defective in mutant mice; if so, this could potentially contribute to the defective plasma leptin response.

This is an excellent point, and we agree that insulin is a potent regulator of leptin production which fluctuates based on the nutritional status, and that changes in fasting insulin might drive and underlie changes in leptin levels. As such, we evaluated plasma insulin but did not observe any significant changes between groups in both fed and fasting conditions. We added the new insulin data to Figure 1.

4) Since leptin action in the brain reverses hyperglucagonemia in rodent models of uncontrolled diabetes, it is possible that deficient leptin action on Pomc cells raises plasma glucagon levels, which in turn increases hepatic glucose production. Measuring glucagon levels would help resolve this possibility.

The reviewers raise an important point that glucagon levels might explain the difference in hepatic glucose production. We did not see any difference in glucagon levels between groups in either fed or fasted conditions. We also performed a glucagon stimulation test and did not observe any differences in the ability of glucagon to stimulate glucose levels. Furthermore, as we noted above in point 3, insulin was not altered in *Pomc*^CreERT2^::*Lepr*^flox/flox^ mice. Taken together, these findings suggest that the effects of POMC neurons on fasting-induced falls in leptin or hepatic glucose production are unlikely to involved changes in pancreatic hormones. Many reports have shown alterations in liver metabolism (lipogenesis, glucose production, insulin sensitivity) following manipulations in POMC neurons. Moreover, a recent report indicates that LEPRs in POMC neurons are required for hepatic parasympathetic nerve responses to leptin (Bell et al., Molecular Metabolism, 2018), which further supports a direct link between POMC neurons and the liver. We now included a discussion of these possibilities in the manuscript. We also added the new glucagon data to Figure 1.

Reviewer #2:[…] Nevertheless, to further improve the manuscript, the authors should include experiments to validate the efficiency and specificity of Lepr-deletion in POMC-neurons in their current model, since this will be of critical importance to interpret the otherwise very interesting set of results.

We thank the reviewer for the very positive assessment of our work. We addressed the specific concern about the validation (see Major issue 1 and Figure 1—figure supplement 1).

Reviewer #4:[…] 1) The authors do not offer a cogent, physiologically-viable explanation for how it is possible for deficient leptin input to Pomc neurons to block the effect of fasting on leptin secretion. The problem is that fasting rapidly and potently lowers plasma leptin levels, presumably reducing leptin action on Pomc cells as a result. Thus, fasting and the mutant mice are each characterized by reduced leptin action in Pomc cells, and yet they have opposite effects on plasma leptin levels. Some effort to bring clarity to this paradox is needed.

We apologize for the lack of clarity, and have extended the Discussion to better explain this apparent paradox.

2) The glucose metabolic phenotype is 1) rather modest, 2) confirmatory of previous literature on Pomc dysfunction, and 3) incomplete in the sense that measures of basal, glucose-stimulated and fasting insulin levels were not included. One wonders, for example, whether the effect of fasting to lower insulin levels is also defective in mutant mice; if so, this could potentially contribute to the defective plasma leptin response.

The points raised are valid concerns. However, we would like to emphasize that blood glucose levels are altered as early as two weeks following the deletion of LEPRs in adult POMC neurons. Importantly, fed and fasting glycemia are 20% and 27% higher four weeks after the deletion, without any change in body weight, which we feel is quite striking. Moreover, the ability of insulin to suppress hepatic glucose production is completely blunted only one week following the deletion, which again is very impressive. While we agree that the results are confirmatory in the sense that they report alterations in glucose metabolism following POMC manipulations, this is the first study that shows that those effects are dissociable from difference in fat mass. We now provide the insulin data from both fed and fasting conditions (see Major issue 3 and Figure 1), and show that insulin does not explain the observed effects.

3) While the clamp data support the conclusion that mutant mice have liver insulin resistance, in the sense that insulin did not inhibit HGP normally, such an effect could potentially involve mechanisms independent of insulin action per se. For example, increased SNS tone to liver and/or islet could increase hepatocyte PKA signaling, which in turn would activate both glycogenolysis and gluconeogenesis. This could occur via either direct (e.g., catecholamine release within the liver) or indirect mechanisms (e.g., increased glucagon secretion). Since leptin action in the brain reverses hyperglucagonemia in rodent models of uncontrolled diabetes (Meek, et al., Endocrinology 2013 Sep;154(9):3067-76), it is possible deficient leptin action on Pomc cells raises plasma glucagon levels, which in turn increases HGP.

We agree with the reviewer that it is important to investigate the role of gluconeogenesis. As suggested, we evaluated fed and fasting glucagon and we also performed a glucagon stimulation test. We did not observe any differences between the groups, suggesting that alterations in glucagon are unlikely to explain our phenotype (see Major issue 4, Figure 1 and Figure 1—figure supplement 2).

4) While the data support the conclusion that leptin receptor signaling in Pomc neurons is required for the effect of fasting to lower leptin levels, they do not exclude a comparable role for leptin signaling in other neurons. Since the authors are likely to have other neuron-specific leptin receptor mutants available, it would be useful to survey them to determine whether the observed phenotype is specific to leptin receptor deletion in Pomc cells.

We agree and appreciate the suggestion that other neurons might be important. We will definitely keep it in mind for future studies.

5) Validation of the mutant phenotype is needed. At a minimum, the authors should test whether following tamoxifen, Pomc neurons no longer respond to leptin in terms of pSTAT3 and/or c-Fos induction.

We have now provided validation data (see Major issue 1 and Figure 1—figure supplement 1).

Reviewer #5:[…] 1) The authors show (Figure 3) that Npy and AgRP mRNA levels are significantly reduced after removing LEPR from POMC neurons, but feeding is normal. What is the interpretation of these data? Do the results mean that NPY and AgRP do not regulate feeding behavior after a fast?

We agree that these findings are very intriguing and believe that much more work needs to be conducted before anyone can have a complete understanding of the observed phenomenon. We believe our observations do not invalidate the importance of NPY/AgRP neurons in regulating feeding behavior after a fast, but do raise important questions about their roles in contexts where leptin levels do not fall adequately. The fact that leptin levels do not fall in fasted *Pomc*^CreERT2^::*Lepr*^flox/flox^ mice might prevent the complete activation of NPY/AgRP neurons, which would place LEPRs-expressing POMC neurons as “gatekeepers” of the leptin/feeding response. Furthermore, the fact that we found no differences in food intake suggests that LEPRs-expressing POMC neurons do not regulate feeding. However, it is surprising that these mice eat the same amount of food despite altered *Npy/Agrp* expression. We now discuss these issues in more details, and hope this will open up a new area of investigation.

2) The extent of Lepr gene inactivation in POMC neurons by tamoxifen treatment should be measured (e.g. loss of leptin-induced pStat3). Since Npy and Agrp mRNA levels are altered, it is also important to show that loss of Lepr is restricted to POMC neurons and not the AgRP/NPY neurons.

We addressed the specific concern about the validation (see Major issue 1 and Figure 1—figure supplement 1).